# An Ex Vivo 3D Tumor Microenvironment-Mimicry Culture to Study TAM Modulation of Cancer Immunotherapy

**DOI:** 10.3390/cells11091583

**Published:** 2022-05-08

**Authors:** Yan-Ruide Li, Yanqi Yu, Adam Kramer, Ryan Hon, Matthew Wilson, James Brown, Lili Yang

**Affiliations:** 1Department of Microbiology, Immunology & Molecular Genetics, University of California, Los Angeles, CA 90095, USA; shuaide@g.ucla.edu (Y.-R.L.); yu63@g.ucla.edu (Y.Y.); akramer931@g.ucla.edu (A.K.); ryanhonchenghao@gmail.com (R.H.); mwilson193@g.ucla.edu (M.W.); brownjimw0@gmail.com (J.B.); 2Eli and Edythe Broad Center of Regenerative Medicine and Stem Cell Research, University of California, Los Angeles, CA 90095, USA; 3Jonsson Comprehensive Cancer Center, David Geffen School of Medicine, University of California, Los Angeles, CA 90095, USA; 4Molecular Biology Institute, University of California, Los Angeles, CA 90095, USA

**Keywords:** tumor-associated macrophage (TAM), tumor microenvironment (TME), ex vivo 3D TME-mimicry culture, chimeric antigen receptor (CAR), CAR-engineered T (CAR-T) cell, cancer immunotherapy, checkpoint inhibitor blockade, monoamine oxidase A (MAO-A) blockade

## Abstract

Tumor-associated macrophages (TAMs) accumulate in the solid tumor microenvironment (TME) and have been shown to promote tumor growth and dampen antitumor immune responses. TAM-mediated suppression of T-cell antitumor reactivity is considered to be a major obstacle for many immunotherapies, including immune checkpoint blockade and adoptive T/CAR-T-cell therapies. An ex vivo culture system closely mimicking the TME can greatly facilitate the study of cancer immunotherapies. Here, we report the development of a 3D TME-mimicry culture that is comprised of the three major components of a human TME, including human tumor cells, TAMs, and tumor antigen-specific T cells. This TME-mimicry culture can readout the TAM-mediated suppression of T-cell antitumor reactivity, and therefore can be used to study TAM modulation of T-cell-based cancer immunotherapy. As a proof-of-principle, the studies of a PD-1/PD-L1 blockade therapy and a MAO-A blockade therapy were performed and validated.

## 1. Introduction

In many solid tumors, tumor-associated macrophages (TAMs) are one of the major components of the tumor microenvironment (TME), and their infiltration and accumulation are strongly associated with poor prognosis in a broad range of solid tumor types [1,2,3,4,5]. TAMs, as well as other immunosuppressive cells such as myeloid-derived suppressor cells (MDSCs) and T regulatory cells, express inhibitory molecules on their cell surfaces and secrete extracellular matrix components, growth factors, cytokines, chemokines, proteases, and metabolites, all of which contribute to the establishment of a hostile and immunosuppressive environment [6,7,8]. Further, TAMs can suppress T-cell-mediated antitumor immunity by releasing IL-10 and TGF-β, amino acid-depleting enzymes such as arginase which cause metabolic starvation on T cells, and by upregulating immune checkpoint ligands such as programmed death-ligand 1 and 2 (PD-L1 and PD-L2) [7,8].

TAM-mediated suppression of T-cell antitumor function is considered to be a major obstacle for many immunotherapies, including immune checkpoint blockade and adoptive T-cell therapies [9,10,11,12,13]. For example, chimeric antigen receptor (CAR)-engineered T (CAR-T) cell therapy has proven to be extremely effective in the treatment of hematological malignancies; however, recent studies have identified the TME as a major contributing factor limiting the efficacy of CAR-T cell therapy to solid tumors [14,15,16,17,18]. Based on these findings, targeting TAMs is a necessary strategy for solid tumor therapeutic intervention. Main therapeutic strategies include clearing and inactivating TAMs by targeting CSF-1/CSF-1R signaling, enhancing macrophage phagocytic activity by blocking CD47-SIRP-α signaling, restricting monocyte recruitment by targeting CCL2R, and inhibiting TAM/T-cell recognition via checkpoint blockade [1,19,20,21,22,23,24].

Although targeting TAMs is a very promising direction, a well-established model to study the therapeutic potential for TAM modulation is still limited. Animal models are a realistic platform for modeling the disease as a whole, but their inherent complexity makes analyzing the contributions of individual cell types problematic [25]. Ex vivo tumor modeling has long supported the discovery of fundamental mechanisms of carcinogenesis and tumor progression [26]. Thus, to devise TAM-targeting therapies, ex vivo models of the TME need to be developed and validated.

In this study, we established an ex vivo 3D TME-mimicry culture comprised of the three major components of a human TME, including human tumor cells, TAMs, and tumor antigen-specific T cells; by using this culture, we were able to investigate TAM immunosuppression on T cells and evaluate the potential of therapeutic candidates for TME modulation. Two macrophage-targeting immunotherapies, immune checkpoint blockade and monoamine oxidase A inhibition, were performed in the ex vivo 3D TME-mimicry cultures and demonstrated antagonizing effects on TAM-suppression of T-cell antitumor reactivity. The application of this culture can be extended to include a large array of molecule-based immunotherapies (e.g., RORγt agonist-, chemokine receptor antagonist-, and toll-like receptor agonist-based therapies), as well as cell-based immunotherapies (e.g., NK, iNKT, and MAIT-cell-based therapies). The culture can also be utilized to study other TME cellular components, such as MDSCs, cancer-associated fibroblasts (CAFs), and T regulatory cells.

## 2. Materials and Methods

### 2.1. Antibodies and Flow Cytometry

All flow cytometry stains were performed in phosphate buffered saline (PBS) for 15 min at 4 °C. The samples were stained with Fixable Viability Dye eFluor506 (e506) mixed with Mouse Fc Block (anti-mouse CD16/32) or Human Fc Receptor Blocking Solution (TrueStain FcX) prior to antibody staining. Fluorochrome-conjugated antibodies specific for human CD45 (Clone H130, PerCP-conjugated, 1:5000), TCRαβ (Clone I26, Pacific blue-conjugated, 1:50), CD3 (Clone OKT3, Pacific blue or PE-conjugated, 1:500), CD4 (Clone OKT4, Pacific blue-conjugated, 1:400), CD8 (Clone SK1, FITC or APC-Cy7-conjugated, 1:400), CD14 (Clone HCD14, Pacific blue-conjugated, 1:1000), CD11b (Clone ICRF44, FITC-conjugated, 1:10,000), CD11c (Clone N418, PerCP-conjugated, 1:1000), CD1d (Clone 51.1, APC-conjugated, 1:50), CD206 (Clone 15-2, APC-conjugated, 1:500), PD-L1 (Clone 10F.9G2, PE-conjugated, 1:2000), PD-1 (Clone 29F.1A12, APC-conjugated, 1:50), CD25 (Clone 3C7, PE-conjugated, 1:2000), CD163 (Clone RM3/1, APC-Cy7-conjugated, 1:500), PD-L2 (Clone 24F.10C12, APC-conjugated, 1:100), HLA-A2 (Clone BB7.2, APC-conjugated, 1:2000), BCMA (Clone 19F2, PE-conjugated, 1:100), Streptavidin (Clone 3A20.2, 1:1000), Granzyme B (Clone QA16A02, APC-conjugated, 1:5000), and Perforin (Clone dG9, PE-Cy7-conjugated, 1:25) were purchased from BioLegend (San Diego, CA, USA). Fluorochrome-conjugated antibody specific for human MSLN (Clone 420,411 PE-conjugated, 1:20) was purchased from R&D Systems (Minneapolis, MN, USA). Goat anti-mouse IgG F(ab’) 2 secondary antibody (1:100) was purchase from Invitrogen (Waltham, MA, USA). Human Fc Receptor Blocking Solution (TrueStain FcX; 1:100) was purchased from Biolegend, and Mouse Fc Block (anti-mouse CD16/32; 1:50) was purchased from BD Biosciences (San Jose, CA, USA). Fixable Viability Dye e506 (1:500) were purchased from Affymetrix eBioscience (San Diego, CA, USA). Intracellular cytokines were stained using a Cell Fixation/Permeabilization Kit (BD Biosciences). Stained cells were analyzed using a MACSQuant Analyzer 10 flow cytometer (Miltenyi Biotech, Auburn, CA, USA). FlowJo software 9 was utilized to analyze the data.

### 2.2. Enzyme-Linked Immunosorbent Cytokine Assays (ELISA)

The ELISAs for detecting human cytokines were performed following a standard protocol from BD Biosciences. Supernatants from co-culture assays were collected and assayed to quantify IFN-γ and TNF-α. The capture and biotinylated pairs for detecting cytokines were purchased from BD Biosciences. The streptavidin-HRP conjugate was purchased from Invitrogen. Human cytokine standards were purchased from eBioscience. Tetramethylbenzidine (TMB) substrate was purchased from KPL (Gaithersburg, MD, USA). The samples were analyzed for absorbance at 450 nm using an Infinite M1000 microplate reader (Tecan, Morrisville, NC, USA).

### 2.3. Lentiviral Vectors

Lentiviral vectors used in this study were all constructed from a parental lentivector pMNDW as previously described [27,28]. The Lenti/BCAR vector was constructed by inserting into pMNDW vector a synthetic gene encoding human BCMA scFV-41BB-CD3ζ-P2A-tEGFR; the Lenti/MCAR vector was constructed by inserting a synthetic gene encoding human αmeso scFV-CD28-41BB-CD3ζ; the Lenti/ESO-TCR vector was constructed by inserting a synthetic gene encoding an HLA-A2-restricted, NY-ESO-1 tumor antigen-specific human CD8 TCR; the Lenti/FG vector was constructed by inserting a synthetic gene encoding Fluc-P2A-EGFP; the Lenti/HLA-A2 vector was constructed by inserting a synthetic gene encoding human HLA-A2.1; and the Lenti/NY-ESO-1 vector was constructed by inserting a synthetic gene encoding human NY-ESO-1. The synthetic gene fragments were obtained from GenScript (Piscataway, NJ, USA) and IDT (Coralville, IA, USA). Lentiviruses were produced using HEK293T cells, following a standard calcium precipitation protocol and an ultracentrifugation concentration protocol as previously described [1,2]. Lentivector titers were measured by transducing HT29 cells with serial dilutions and performing digital qPCR, following established protocols [27,28].

### 2.4. Human Monocyte-Derived Macrophage (MDM) Culture and Polarization

Healthy donor peripheral blood mononuclear cells (PBMCs) were obtained from the CFAR Gene and Cellular Therapy Core Laboratory at UCLA, without identification information under federal and state regulations. Human monocytes were isolated from PBMCs by adherence. Briefly, PBMCs were suspended in serum-free RPMI 1640 media (Corning Cellgro, Manassas, VA, USA, #10-040-CV) at 1 × 10^7^ cells/mL. About 10–15 mL of the cell suspension were added to each 10 cm dish and incubated for an hour in a humidified 37 °C, 5% CO_2_ incubator. Next, medium containing non-adherent cells was discarded. The dishes were then washed twice using PBS, and the adherent monocytes were cultured in C10 medium supplemented with human M-CSF (10 ng/mL) (Peprotech, Rocky Hill, NJ, USA, #300-25) for 6 days to generate MDMs. At day 6, the generated MDMs were dissociated by 0.25% Trypsin/EDTA (Gibco, Waltham, MA, USA, #25200-056), collected, and reseeded in a 6 or 12-well plate in C10 medium (0.5–1 × 10^6^ cells/mL) for 48 hours in the presence of recombinant human IL-4 (10 ng/mL) (Peprotech, #214-14) and human IL-13 (10 ng/mL) (Peprotech, #214-13) to induce MDM polarization. In some experiments, MAO-A inhibitor (phenelzine, 20 μM) was added to the MDM polarization culture 2–4 days prior to adding recombinant human IL-4 and human IL-13, to block MAO-A activity during MDM polarization. Polarized MDMs were then collected and used for flow cytometry or for setting up in vitro mixed culture experiments.

### 2.5. Generation of Human BCMA CAR-Engineered T (BCAR-T) Cells, Mesothelin CAR-Engineered T (MCAR-T) Cells, and NY-ESO-1-Specific TCR-Engineered T (ESO-T) Cells

Healthy donor PBMCs were cultured in a 12-well plate in C10 medium (1 × 10^6^ cells/mL/well) for 2 days and stimulated with Dynabeads™ Human T-Activator CD3/CD28 (10 μL/mL) (GIBCO, 11161D) and recombinant human IL-2 (20 ng/mL) (Peprotech). After 2 days, Dynabeads™ were removed and cells were spin-infected with frozen-thawed Lenti/ESO-TCR, Lenti/BCAR, or Lenti/MCAR viruses supplemented with polybrene (10 μg/mL) at 660× *g* at 30 °C for 90 min following an established protocol. Virus-transduced T cells were expanded for another 6–8 days in C10 medium supplemented with recombinant human IL-2 (20 ng/mL) (Peprotech) and cryopreserved for future use. CAR or ESO-TCR expression levels on T cells were determined using flow cytometry.

### 2.6. In Vitro Mixed Mφ/T-Cell Reaction Assay

Healthy donor PBMCs were collected and co-cultured with autologous M2-polarized macrophages at a 1:1 ratio in 96-well round bottom plates in C10 medium for 3 days. Cell numbers were indicated in figure legends. PBMCs collected from multiple donors were studied. Human anti-CD3 (clone HIT3a, Biolegend) and anti-CD28 (clone CD28.2, Biolegend) antibodies were added to the culture to stimulate PBMC-T-cell expansion. The cells were collected to study surface marker expression and intracellular cytotoxicity molecule production using flow cytometry, and the cell culture supernatants were collected to measure cytokine production using ELISA.

### 2.7. In Vitro Tumor Cell Killing Assay

Tumor cells (1 × 10^4^ cells per well) were co-cultured with effector cells (at ratios indicated in figure legends) in Corning 96-well clear bottom black plates for 24 h, in C10 medium. At the end of culture, live tumor cells were quantified by adding D-luciferin (150 mg/mL; Caliper Life Science, Hopkinton, MA, USA) to cell cultures and reading out luciferase activities using an Infinite M1000 microplate reader (Tecan).

### 2.8. Ex Vivo 3D TME-Mimicry Culture

The tumor cells, M2 macrophages, and autologous CAR-T or ESO-T cells were collected and mixed at ratios indicated in figure legends. Mixed cells were centrifuged and resuspended in C10 medium at 1 × 10^5^ cells per μL medium. The cell slurry was adjusted to 5–10 μL per aggregate and was gently transferred onto a microporous membrane cell insert (EMD Millipore, Billerica, MA, USA, #PICM0RG50) using a 20 μL pipet to form a 3D human tumor/TAM/T-cell organoid. Prior to cell transfer, cell inserts were placed in a six-well plate immersed with 1 mL C10 medium. Two days later, the organoids were dissociated by P1000 pipet tip and disrupted through a 70-μm nylon strainer to generate single-cell suspensions for further analysis. In some experiments, 50 μg/mL of anti-human PD-L1 antibody (B7-H1; clone 29E.2A3, BioXCell, Lebanon, NH, USA) or Mouse IgG2b, κ control antibody was added to the organoid to study the effect of anti-PD-L1 antibody to TAM-mediated T-cell antitumor reactivity suppression.

## 3. Results

### 3.1. Generation of Human Monocyte-Derived M2-Polarized Macrophages

Human peripheral blood mononuclear cell (PBMC) monocyte-derived macrophages (MDMs) were cultured in vitro and polarized toward an immunosuppressive phenotype (denoted as M2 macrophage or M2 Mφ) by adding anti-inflammatory stimuli (i.e., IL-4 and IL-13; Figure 1A,B) [29]. Compared to PBMC-derived monocytes, M2 macrophages expressed higher levels of M2 macrophage markers (i.e., CD11b, CD206, CD163) and immune checkpoint receptor-ligands (i.e., PD-L1 and PD-L2; Figure 1B,C). The capacity to release anti-inflammatory cytokines (e.g., IL-10) and the engagement of immune checkpoint (e.g., PD-L1) are the signatures of M2 macrophages in triggering inflammation resolution and suppression of T-cell activation [30].

### 3.2. Validation of M2-Polarized Macrophage Immunosuppressive Function

To study the M2 macrophage-mediated PBMC-T-cell suppression, we performed an in vitro mixed macrophage/PBMC-T-cell (Mφ/T-cell) reaction assay (Figure 1D). M2 macrophages significantly suppressed CD4^+^ T and CD8^+^ T-cell expansion and activation, evidenced by reduced T-cell numbers (Figure 1E), down-regulated T-cell activation marker CD25 (Figure 1F,G), down-regulated cytotoxicity molecules Perforin and Granzyme B (Figure 1F,G), and decreased T-cell-secreted cytokines IFN-γ and TNF-α (Figure 1H). Interestingly, M2 macrophages induced the downregulation of PD-1 on T cells, coinciding with the upregulation of PD-L1 on macrophages (Figure 1I,J), indicating the engagement of PD-1/PD-L1 immune checkpoints between PBMC-T cells and autologous M2 macrophages.

### 3.3. Validation of PD-1/PD-L1 Blockade on Antagonizing M2 Macrophage-Mediated Immunosuppression

Immune checkpoint inhibitors, such as anti-programmed cell death protein 1 (PD-1), anti-programmed cell death ligand 1 (PD-L1), and anti-cytotoxic T-lymphocyte-associated protein 4 (CTLA4), have been widely used in anti-cancer immunotherapies and resulted in significant improvements in disease outcome for a variety of cancers [31]. Here we used the anti-PD-L1 antibody as an immune checkpoint inhibitor representative and performed an in vitro mixed Mφ/T-cell reaction assay to study the effects of immune checkpoint blockade on M2 macrophage-mediated T-cell suppression (Figure 2A). In the mixed reaction assay, macrophages significantly suppressed CD4^+^ T and CD8^+^ T-cell expansion, while the addition of anti-PD-L1 antibody antagonized the macrophage-mediated T-cell suppression, evidenced by rescued T-cell expansion and increased secretion of IFN-γ and TNF-α by T cells (Figure 2B–E). These data indicate the promise of TAM-facing checkpoint blockade for cancer therapy.

### 3.4. Validation of MAO-A Blockade on Antagonizing M2 Macrophage-Mediated Immunosuppression

MAO-A is an outer mitochondrial membrane-bound enzyme [32]. A recent study has shown that MAO-A promotes tumor-associated macrophage (TAM) immunosuppressive polarization and inhibits antitumor immunity in mice by upregulating oxidative stress [32]. MAO inhibitors (e.g., phenelzine) induce TAM reprogramming and suppress tumor growth [32]. Here, we used phenelzine as an MAO-A inhibitor representative and performed an in vitro mixed Mφ/T-cell reaction assay to study the antagonization of MAO-A blockade toward M2 macrophage-mediated T-cell suppression (Figure 2F). Notably, phenelzine significantly inhibited IL-4/IL-13-induced immunosuppressive polarization of MDMs, evidenced by their down-regulated immunosuppressive markers (i.e., CD206) and immune checkpoint receptor-ligands (i.e., PD-L1 and PD-L2; Figure 2G). In the in vitro mixed Mφ/T-cell reaction assay, compared to the PBMC-T cells co-cultured with M2 macrophages, the PBMC-T cells co-cultured with phenelzine-treated M2 macrophages expanded faster (Figure 2H,I), and secreted more pro-inflammatory cytokines (i.e., IFN-γ and TNF-α; Figure 2J,K), indicating rescued T-cell expansion and activation. Therefore, MAO-A blockade could effectively antagonize the immunosuppressive function of M2 macrophages, supporting an attractive potential of MAO-A blockade for TME-targeting cancer immunotherapy.

### 3.5. Development of an Ex Vivo 3D TME-Mimicry Culture to Study TAM Modulation of T-Cell Antitumor Reactivity 

Chimeric antigen receptor (CAR)-engineered T (CAR-T) and antigen-specific TCR-engineered T (TCR-T) cell therapies have demonstrated remarkable efficacy for the treatment of hematological malignancies [33,34,35,36,37]. However, in some patients with solid tumors, objective responses to CAR-T or TCR-T-cell therapy remain sporadic and transient [18]. The immunosuppressive environment of the TME in advanced solid tumors can greatly reduce CAR-T or TCR-T efficacy in solid tumors, positioning the regulatory immune cells of the TME as targets for potential cancer therapy [18].

To study the TAM modulation of CAR-T or TCR-T-cell antitumor reactivity, we used an ex vivo 3D TME-mimicry culture (Figure 3A). Three antigens were chosen as the model tumor antigens: B-cell maturation antigen (BCMA), a specific tumor antigen for multiple myeloma (MM) [38]; Mesothelin (MSLN), a tumor differentiation antigen expressed at low levels in normal tissue but overexpressed in a broad range of solid tumors such as mesothelioma, ovarian cancer, breast cancer, pancreatic cancer, lung cancer, etc., [39]; and NY-ESO-1, a well-recognized tumor antigen commonly expressed in a large variety of human cancers including neuroblastoma, myeloma, metastatic melanoma, ovarian cancer, prostate cancer, breast cancer, etc., [40]. Three tumor cell lines were used as targets, multiple myeloma (MM.1S), ovarian cancer (OVCAR3), and melanoma (A375) cell lines (Figure 3B). All three tumor cell lines were engineered to overexpress the firefly luciferase (Fluc) and enhanced green fluorescence protein (EGFP) dual reporters (denoted as FG) to enable convenient monitoring using flow cytometry or luciferase assay (Appendix A) [1]. A375 melanoma cell line was also engineered to co-express NY-ESO-1 as well as its matching MHC molecule, HLA-A2, to serve as the human tumor target (Appendix A) [28]. BCMA-targeting CAR-T (BCAR-T), MSLN-targeting CAR-T (MCAR-T), and NY-ESO-1-specific TCR-engineered T (ESO-T) cells were generated by transducing healthy donor PBMCs with a Lenti/BCAR lentivector, a Lenti/MCAR lentivector, and a Lenti/ESO-TCR vector, respectively (Figure 3C–F, and Appendix A). Using an in vitro tumor-cell killing assay, we evaluated the tumor killing efficacy of CAR-T or ESO-T cells in comparison with mock-transduced PBMC-T cells (Appendix A). All three BCAR-T, MCAR-T, and ESO-T cells exhibited significantly enhanced tumor killing efficacies toward MM.1S-FG, OVCAR3-FG, and A375-A2-ESO-FG tumor cells, respectively (Appendix A).

The tumor cells, CAR-T or ESO-T cells, and M2-polarized macrophages were mixed at indicated ratios and seeded to form a 3D tumor organoid mimicking the TME. Among the three tumor models, M2-polarized macrophages effectively suppressed CAR-T and ESO-T-cell-mediated killing of tumor cells (Figure 3G,J,M). Accordingly, CAR-T or ESO-T cells co-cultured with M2-polarized macrophages, compared to those that did not co-culture with macrophages, showing a decrease in T-cell activation and cytotoxicity (i.e., decreased CD25 expression, increased CD62L expression, and decreased Granzyme B production; Figure 4H,I,K,L,N,O). Collectively, these results suggest that human TAMs significantly suppress T-cell antitumor reactivity in the ex vivo 3D TME-mimicry culture.

### 3.6. PD-1/PD-L1 Blockade Antagonizes TAM-Suppression of T-Cell Antitumor Reactivity

To study the effect of immune checkpoint blockade on TAM-modulated T-cell antitumor capacity, we set up the ex vivo 3D TME-mimicry culture with the addition of anti-PD-L1 antibody (Figure 4A). OVCAR3-FG human ovarian cancer cell line and MCAR-T cells were chosen for the models (Figure 4A). TAMs effectively suppressed MCAR-T-cell-mediated killing of OVCAR3-FG tumor cells, and this immunosuppressive effect was largely limited by adding anti-PD-L1 antibody (Figure 4B). Accordingly, compared to the MCAR-T cells co-cultured with M2-polarized macrophages only, those cells co-cultured with macrophages and anti-PD-L1 showed a rescued T-cell activation and cytotoxicity, as evidenced by increased production of Granzyme B (Figure 4C), upregulation of CD25 (Figure 4D), and down-regulation of CD62L in MCAR-T cells (Figure 4E). Collectively, these data support the application of the 3D TME-mimicry culture and demonstrate that immune checkpoint inhibitors (i.e., anti-PD-L1) alleviate the immunosuppressive effect of human TAMs, suggesting the potential of immune checkpoint blockade to increase T-cell antitumor capacity.

### 3.7. MAO-A Blockade Antagonizes TAM-Suppression of T-Cell Antitumor Reactivity

To study whether MAO inhibitor-induced human TAM reprogramming could affect human T-cell antitumor reactivity, we used phenelzine-treated macrophages to set up the ex vivo 3D TME-mimicry culture (Figure 5A). OVCAR3-FG human ovarian cancer cell line and MCAR-T cells were chosen for the models (Figure 5A). TAMs effectively suppressed MCAR-T-cell-mediated killing of OVCAR3-FG tumor cells in the absence of MAO inhibitor treatment; however, this immunosuppressive effect was significantly alleviated by phenelzine treatment during MDM polarization (Figure 5B). Accordingly, compared to the MCAR-T cells co-cultured with M2-polarized macrophages, those cells co-cultured with phenelzine-treated macrophages showed a rescued T-cell activation and cytotoxicity, as evidenced by increased production of Granzyme B (Figure 5C), upregulation of CD25 (Figure 5D), and downregulation of CD62L in MCAR-T cells (Figure 5E). Collectively, the ex vivo 3D TME-mimicry culture can be used to study the effect of MAO-A blockade on TAM reprogramming, which has the potential to antagonize TAM immunosuppression and enhance T-cell antitumor responses. Notably, MAO inhibitors such as phenelzine could also upregulate autocrine serotonin signaling in T cells and significantly enhance T-cell antitumor function [41]. Therefore, adding phenelzine into the 3D TME-mimicry culture may potentially improve T-cell antitumor reactivity by directly interacting with T cells. This 3D TME-mimicry culture could be utilized to screen a variety of drugs in order to study their effects on each cell component (i.e., tumor cells, TAM or T cells) and cell interactions in the culture.

## 4. Discussion

The cellular and molecular properties of the TME in modulating tumor progression and metastasis have been greatly emphasized in recent years [3,7,8]. TAMs are the major cells in creating an immunosuppressive TME by producing cytokines, chemokines, and growth factors, and triggering the release of inhibitory immune checkpoint proteins in T cells [6,7]. The behavior of TAMs must be considered when developing new cancer therapies. The goal of this study was to establish an ex vivo system to closely mimic the human TME; by using this ex vivo 3D TME-mimicry culture, we were able to identify the mechanisms driving TAM-mediated suppression of T-cell functions and identify the therapeutic candidates to modulate the TME for cancer immunotherapy.

We first present here an ex vivo 3D culture. The specific aim for establishing a 3D multicellular structure was to facilitate 3D cell–cell interactions and better model the in vivo TME. It has been shown that the spatial cell–cell interactions regulate cancer development, tissue homeostasis, and single-cell functions. When tumor cells cultured in 2D and 3D cultures were compared, significant differences emerged not only in morphology but also in key biological features such as gene and protein expressions, growth rates, and invasive behaviors [42,43,44,45,46]. One benefit of 3D culture is that cells grown in 3D culture more closely resemble those grown in vivo, in terms of their morphology, gene expression, and metabolism. Hepatocytes, for example, when cultured in 3D, adopt in vivo-like morphology, polarity, and the expression of various liver-specific activities [47]. Moreover, the 3D culture promotes the upregulation of proteins involved in cell survival and drug resistance in cancer cells [42]. Therefore, tumor cells from 3D culture were more pharmacologically resistant to anticancer drugs than those from 2D, and were more relevant to in vivo conditions. For instance, 3D cultures of HER2-positive breast cancer cell lines exhibited higher resistance to both chemotherapy and HER-targeted drugs when compared to 2D cultures [42]. 3D cultures of pancreatic cancer cell lines, MIAPaCa-2 and PANC-1, were less sensitive to chemotherapeutic drug 5-fluorouracil in comparison to cells in 2D [48]. Similarly, multiple myeloma cell lines grown in 3D cultures showed higher resistance to both bortezomib and carfilzomib than those grown in 2D [49]. The deviation of pharmacological responses observed between 2D and 3D culture system could be partially responsible for the high failure rate associated with drug development, since most preclinical cell-based screenings are performed in 2D systems. Together, those findings emphasize the necessity of developing 3D culture systems for studying the TME.

Another major motivation of our study was to create a better in vitro TME-mimicry model by incorporating three major components of the human TME: TAMs, tumor cells, and tumor antigen-specific T cells. Homogenous 3D tumor spheroids from a single cell line have been used to assess the antitumor capacity of CAR-T cells [50,51], cytotoxic T cells [52], and NK cells [53] in vitro. However, validation of those cell-based immunotherapies made in 3D spheroid models is often decoupled from in vivo results [50]. One contributing factor is that most of the current 3D in vitro cancer models consist only of cancer cells and are unable to reproduce the complex cellular interactions caused by tumor-associated immune cells within the TME [35]. Thus, it is of significant importance to develop a 3D model incorporating TAMs. Because M2 macrophages make up the majority of TAMs [11], we used in vitro-generated M2 macrophages to model TAM functions in this study. We showed that TAMs effectively suppressed both CAR-T and ESO-T-cell-mediated killing of tumor cells in the ex vivo 3D culture system (Figure 3), as they do in situ. Insights into CAR-T-cell function were also provided through alternative readouts, including activation markers and cytotoxic molecules. We further validated our 3D TME-mimicry culture system by evaluating the antitumor response of two TAM-targeting immunotherapies. We showed that blockage of PD-1/PD-L1 interaction between TAMs and MCAR-T cells pronouncedly restored the antitumor efficacies of the MCAR-T cells (Figure 4), in agreement with both mouse models and clinical response [36,37]. Further, reprogramming the polarization of TAMs using a MAO-A inhibitor significantly reduced the immunosuppressive activities of TAMs and restored antitumor activity of CAR-T cells (Figure 5) in accordance with the results obtained from syngeneic mouse tumor models [25]. We envision that the ex vivo 3D TME-mimicry culture presented here could be further applied to evaluating the therapeutic effects of a large array of molecule-based immunotherapies targeting TAMs including CCL2/CCR2 inhibition [54], CSF1/CSF1R blockade [22], CD40 agonists [55], HDAC inhibition [56], and PI3Kγ inhibition [57]. Furthermore, the application of the 3D TME-mimicry culture can be extended to study the antitumor capacity of other therapeutic cells such as NK, iNKT, MAIT, and γδT cells, providing a valuable tool for cell-based TAM-targeting immunotherapy.

Despite the promise, one possible limitation of the 3D culture system reported here is that it only allows us to elucidate the TAM-mediated modulation in the TME. The TME is made up of a variety of immunoregulatory cells that have been subverted by cancer cells to aid tumor development. To study the suppressive effect of other tumor-associated cells, we propose to incorporate other major components of the TME, including cancer-associated fibroblast, myeloid-derived-suppressor cells, and T regulatory cells into this system [58,59,60]. We envision that this system and future adaptations may benefit mechanistic studies on the role of the TME in cancer immunotherapy and provide a more accurate preclinical ex vivo platform for assessment of potential cancer immunotherapies.

## 5. Conclusions

In summary, we provide here an ex vivo 3D culture that retains key immune features of the native TME for studying the TAM modulation of T-cell functions. Our model captures the relationships between macrophages and T cells within the TME and predicts how TAM-targeting immunotherapies impact the immune response to the tumor. We highlight the importance of modeling the TME in vitro, allowing a deeper understanding of the key molecular and cellular interactions related to cancer immunotherapy.

## Figures and Tables

**Figure 1 cells-11-01583-f001:**
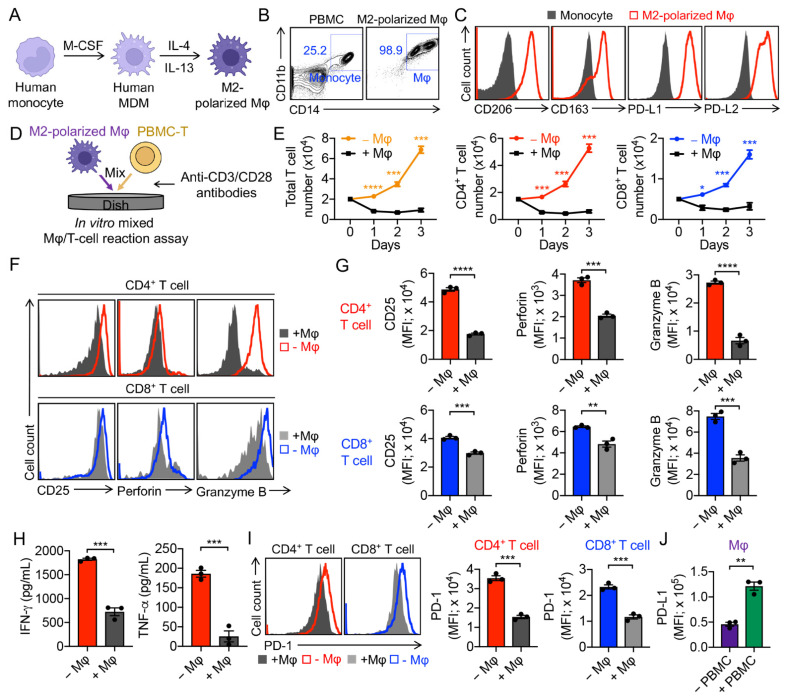
Generation and validation of human monocyte-derived M2-polarized immunosuppressive macrophages. (**A**) Diagram showing the human monocyte-derived M2 macrophage culture and polarization. M-CSF, macrophage colony-stimulating factor; MDM, monocyte-derived macrophage; Mφ, macrophage. (**B**) Fluorescence-activated cell sorting (FACS) detection of CD11b and CD14 on M2-polarized macrophages. Healthy donor peripheral blood mononuclear cells (PBMCs) were included as a staining control. (**C**) FACS detection of surface markers on M2-polarized macrophages. Monocytes were included as a control. (**D**–**J**) In vitro mixed Mφ/T-cell reaction assay to study M2 macrophage-mediated T-cell suppression. (**D**) Experimental design. 1 × 10^5^ PBMCs were cultured in the assay. (**E**) PBMC-T-cell growth curve (*n* = 3). (**F**) FACS detection of surface marker (CD25) and intracellular cytotoxic molecules (Perforin and Granzyme B) of CD4^+^ and CD8^+^ T cells. (**G**) Quantification of F (*n* = 3). (**H**) ELISA analyses of cytokine (IFN-γ and TNF-α) production in the mixed reaction assay at day 3 (*n* = 3). (**I**) FACS analyses of PD-1 on CD4^+^ and CD8^+^ T cells (*n* = 3). (**J**) FACS analyses of PD-L1 on M2 macrophages (*n* = 3). Representative of 3 (**D**–**J**) and > 5 (**A**–**C**) experiments. Data are presented as the mean ± SEM. * *p* < 0.05, ** *p* < 0.01, *** *p* < 0.001, **** *p* < 0.0001, by Student’s *t* test.

**Figure 2 cells-11-01583-f002:**
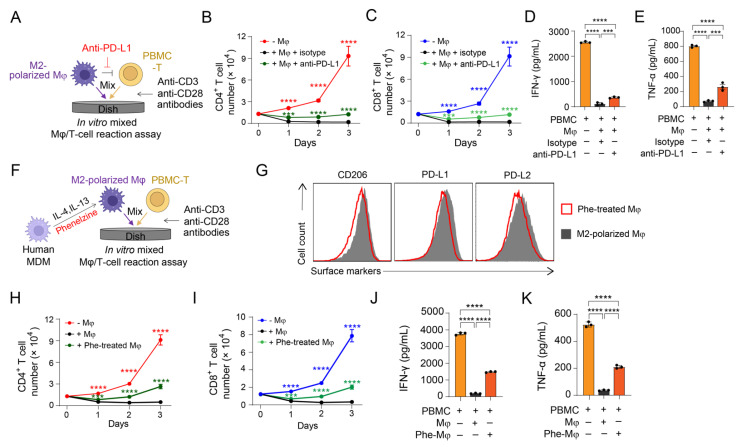
Validation of immune modulatory reagents on antagonizing M2 macrophage-mediated immunosuppression. (**A**–**E**) Study the effect of anti-PD-L1 antibody that blocks macrophage-T-cell inhibitory interaction. (**A**) Experimental design. 1 × 10^5^ PBMCs were cultured in the assay. (**B**) CD4^+^ PBMC-T-cell growth curve (*n* = 3). (**C**) CD8^+^ PBMC-T-cell growth curve (*n* = 3). (**D**,**E**) ELISA analyses of IFN-γ (**D**) and TNF-α (**E**) production in the mixed reaction assay at day 3 (*n* = 3). (**F**–**K**) Study the effect of MAO-A inhibitor phenelzine that reprograms M2 macrophage polarization. (F) Experimental design. About 1 × 10^5^ PBMCs were cultured in the assay. (**G**) FACS detection of CD206, PD-L1, and PD-L2 on phenelzine-treated or non-treated M2-polarized macrophages. Phe, phenelzine. (**H**) CD4^+^ PBMC-T-cell growth curve (*n* = 3). (**I**) CD8^+^ PBMC-T-cell growth curve (*n* = 3). (**J**,**K**) ELISA analyses of IFN-γ (**J**) and TNF-α (**K**) production in the mixed reaction assay at day 3 (*n* = 3). Representative of three experiments. Data are presented as the mean ± SEM. ns, not significant, *** *p* < 0.001, **** *p* < 0.0001, by one-way ANOVA.

**Figure 3 cells-11-01583-f003:**
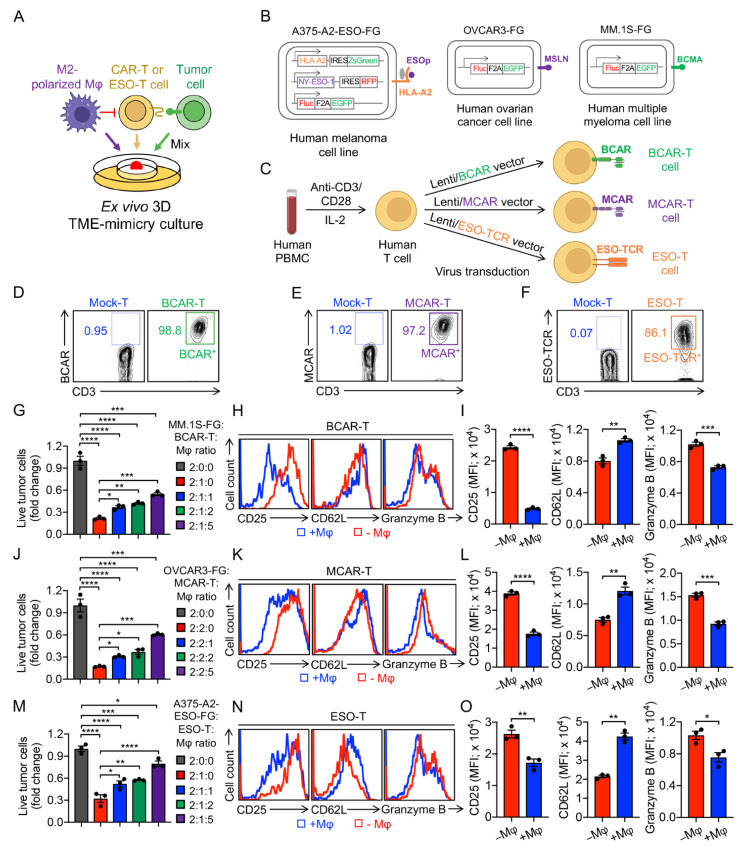
Development of an ex vivo 3D TME-mimicry culture to study TAM modulation of T-cell antitumor reactivity. (**A**) Diagram of an ex vivo 3D TME-mimicry culture. Three human tumor cell lines were studied: MM.1S (multiple myeloma), OVCAR3 (ovarian), and A375 (melanoma). (**B**) Schematics showing the engineered MM.1S-FG, OVCAR3-FG, and A375-A2-ESO-FG cell lines. Fluc, firefly luciferase; EGFP, enhanced green fluorescent protein; FG, Fluc-EGFP; F2A, foot-and-mouth disease virus 2A; RFP, red fluorescent protein; NY-ESO-1, New York esophageal squamous cell carcinoma-1; ESOp, ESO peptide; IRES, internal ribosome entry site; HLA, human leukocyte antigen. (**C**–**F**) Generation of BCMA CAR-engineered T (BCAR-T), mesothelin CAR-engineered T (MCAR-T), and HLA-A2-restricted, NY-ESO-1 tumor antigen-specific human CD8 TCR-engineered T (ESO-T) cells. (**C**) Experimental design. (**D**–**F**) FACS detection of BCAR on BCAR-T cells (**D**), MCAR on MCAR-T cells (**E**), and ESO-TCR on ESO-T cells (**F**). Human T cells that received mock transduction were included as a staining control (denoted as Mock-T). (**G**–**O**) TAM modulation of T-cell antitumor reactivity. (**G**) Tumor killing data of MM.1S-FG by BCAR-T cells at 48 h (*n* = 3). (**H**) FACS detection of surface markers (CD25 and CD62L) and intracellular cytotoxic molecule (Granzyme B) of BCAR-T cells (**I**) Quantification of H (*n* = 3). (**J**) Tumor killing data of OVCAR3-FG by MCAR-T cells at 48 h (*n* = 3). (**H**) FACS detection of surface markers and intracellular cytotoxic molecule by MCAR-T cells (**L**) Quantification of K (*n* = 3). (**M**) Tumor killing data of A375-A2-ESO-FG by ESO-T cells at 48 h (*n* = 3). (**N**) FACS detection of surface markers and intracellular cytotoxic molecule by ESO-T cells. (**O**) Quantification of N (*n* = 3). Representative of three experiments. Data are presented as the mean ± SEM. * *p* < 0.05, ** *p* < 0.01, *** *p* < 0.001, **** *p* < 0.0001, by Student’s t test (**I**,**L**,**O**), or by one-way ANOVA (**G**,**J**,**M**).

**Figure 4 cells-11-01583-f004:**
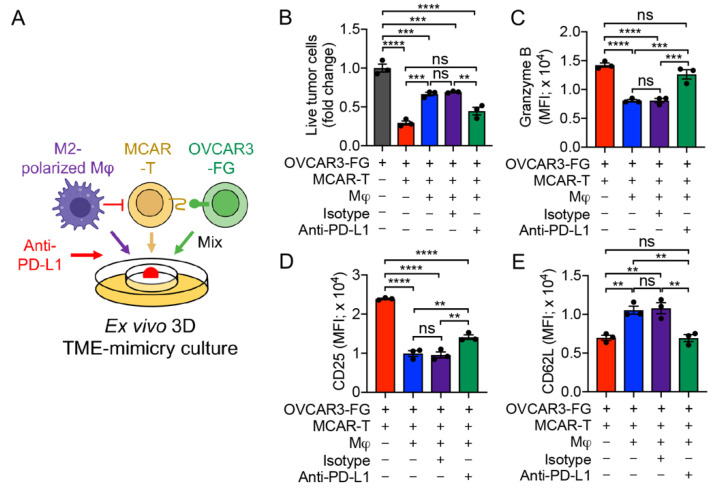
Application of the ex vivo 3D TME-mimicry culture: studying the PD-1/PD-L1 blockade therapy. (**A**) Experimental design. OVCAR-3-FG and MCAR-T cells were studied. (**B**) Tumor killing data at 48 h (*n* = 3). (**C**–**E**) FACS analyses of intracellular cytotoxicity molecule Granzyme B (**C**), and surface marker CD25 (**D**) and CD62L (**E**) of MCAR-T cells (*n* = 3). Representative of three experiments. Data are presented as the mean ± SEM. ns, not significant, ** *p* < 0.01, *** *p* < 0.001, **** *p* < 0.0001, by one-way ANOVA.

**Figure 5 cells-11-01583-f005:**
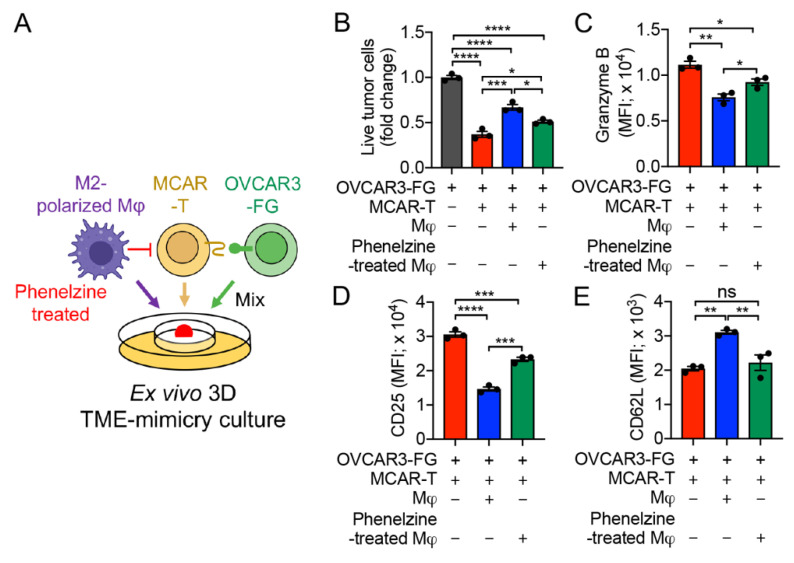
Application of the ex vivo 3D TME-mimicry culture: studying the MAO-A blockade therapy. (**A**) Experimental design. OVCAR-3-FG and MCAR-T cells were studied. (**B**) Tumor killing data at 48 h (*n* = 3). (**C**–**E**) FACS analyses of intracellular cytotoxicity molecule Granzyme B (**C**), and surface marker CD25 (**D**) and CD62L (**E**) of MCAR-T cells (*n* = 3). Representative of three experiments. Data are presented as the mean ± SEM. ns, not significant, * *p* < 0.05, ** *p* < 0.01, *** *p* < 0.001, **** *p* < 0.0001, by one-way ANOVA.

## Data Availability

Data supporting reported results are available on request from the corresponding author.

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
