# Peer review of "An Ex Vivo 3D Tumor Microenvironment-Mimicry Culture to Study TAM Modulation of Cancer Immunotherapy"

_cells, 2022, doi:10.3390/cells11091583_

Round 1

Reviewer 1 Report

The manuscript is well written with figures clearly presented. The results are interesting provide a new direction for screening drugs that could modulate immunotherapy.

It would add strength if the MOA inhibitor addition to the 3D culture directly instead of treating m2 macrophages separately.  how does it alter T cell response in vivo?

minor concern regarding FCAS staining: in general, FACS buffer contains serum of BSA so it helps cells prevent clumping and non-specific binding of antibodies. why was PBS used in this study for FACS staining?

Author Response

The manuscript is well written with figures clearly presented. The results are interesting provide a new direction for screening drugs that could modulate immunotherapy.

It would add strength if the MAO inhibitor addition to the 3D culture directly instead of treating m2 macrophages separately. how does it alter T cell response in vivo?

Response: We appreciate the Reviewer’s constructive comment.

MAO inhibitors such as Phenelzine could significantly upregulate serotonin signaling in T cells and enhance T cell antitumor function (Wang et al., Sci. Immunol. 6, eabh2383). In our current study, we mainly focused on the TAM/T cell interaction (Figure 4) and TAM polarization reprogramming (Figure 5). Directly adding MAO inhibitor to the 3D culture will strengthen the study, however, it’s more complicated to draw a conclusion about which cell types (i.e., TAM and T cells) the MAO inhibitor will affect in the 3D culture, because MAO inhibitor will interact with both TAM and T cells. Nevertheless, we will discuss the feasibility of studying the T cell/MAO inhibitor interaction in the manuscript. We added “Notably, MAO inhibitors such as phenelzine could also upregulate autocrine serotonin signaling in T cells and significantly enhance T cell antitumor function [41]. Therefore, adding phenelzine into the 3D TME-mimicry culture could potentially improve T cell antitumor reactivity by directly interacting with T cells. This 3D TME-mimicry culture can be utilized to screen a variety of drugs in order to study their effects on each cell component (i.e., tumor cells, TAM or T cells) and cell interactions in the culture.” in Line 363-369.

minor concern regarding FCAS staining: in general, FACS buffer contains serum of BSA so it helps cells prevent clumping and non-specific binding of antibodies. why was PBS used in this study for FACS staining?

Response: We thank the Reviewer for pointing this out. We agree with that FACS buffer containing BSA helps cells preventing clumping and non-specific staining. In our protocol, we used Mouse Fc Block (anti-mouse CD16/32) or Human Fc Receptor Blocking Solution (TrueStain FcX) prior to antibody staining, to prevent the non-specific staining. In addition, the flow cytometry stains were performed for 15 min at 4 °C, and cells did not clump in such a short time. If necessary, cells could be filtered to single cell suspension before running in the MACSQuant Analyzer 10 flow cytometer. We revised our Methods Section and described it more clearly.

Reviewer 2 Report

Authors report the development of an ex vivo 3D TME-mimicry culture comprising three major components of human TME, including human tumor cells, TAMs, and tumor antigen-specific T cells. In the manuscript, authors show the most of data using flow cytometry, which depends on the specificity of antibodies. data and conclusion are understandable, however provide the data showing the specificity of each antibody, by such as westernblot.

In Fig. 1D, experimental design, and data (1E)

How much of cell numbers were mixed? If same cell numbers mixed in Fig. 1E and Fig. 2BC, without MΦ should show similar data by CD4+ and CD8+ T-cell numbers at day 3. Authors provide detail explanation.

In Fig. 2D, authors should add the data PBMC(+) and MΦ(+) without anti-PD-L1.

In Fig. 3BC, authors should check the expression of each protein by westernblot.

Provide detail information in method section.

Check all of the references.

Round 2

Reviewer 2 Report

Authors mentioned that PBMC-T cells from different donors had different CD4/CD8 ratio and exhibited different proliferation capacity. If this is true, authors should confirm the expression of each protein by each antibody in the experiment. Before using antibody, authors should confirm the specificity of each antibody. Based on our knowledge, some of the antibodies are not reliable, even if the company verified the specificity. The authors over-reliance on the product information presented by each company.

Author Response

We really appreciate the Reviewer’s inputs. However, it’s difficult to confirm the specificity of each antibody, because we used over 30 different antibodies in our study, and not all antibodies that are functional by flow cytometry are suitable by westerns. We consulted with the academic editor from Cells journal, and the editor agreed to waive the western blot experiments but providing exact dilution of the flow antibodies and further elaboration on flow cytometry protocol. In the Materials and Methods section, we provided the conjugated fluorochrome and exact dilution of all antibodies, and we modified the flow cytometry protocol (Line 77-108).